# Why Polyurethanes Have Been Used in the Manufacture and Design of Cardiovascular Devices: A Systematic Review

**DOI:** 10.3390/ma13153250

**Published:** 2020-07-22

**Authors:** Kelly Navas-Gómez, Manuel F. Valero

**Affiliations:** Energy, Materials and Environment Group, Faculty of Engineering, Universidad de La Sabana, Chía 140013, Colombia; kelly.navas@unisabana.edu.co

**Keywords:** polyurethanes, cardiovascular devices, polyurethanes properties, functionalization, systematic review

## Abstract

We conducted a systematic review in accordance with the Preferred Reporting Items for Systematic Reviews and Meta-Analyses (PRISMA) statement to ascertain why polyurethanes (PUs) have been used in the manufacture and design of cardiovascular devices. A complete database search was performed with PubMed, Scopus, and Web of Science as the information sources. The search period ranged from 1 January 2005 to 31 December 2019. We recovered 1552 articles in the first stage. After the duplicate selection and extraction procedures, a total of 21 papers were included in the analysis. We concluded that polyurethanes are being applied in medical devices because they have the capability to tolerate contractile forces that originate during the cardiac cycle without undergoing plastic deformation or failure, and the capability to imitate the behaviors of different tissues. Studies have reported that polyurethanes cause severe problems when applied in blood-contacting devices that are implanted for long periods. However, the chemical compositions and surface characteristics of polyurethanes can be modified to improve their mechanical properties, blood compatibility, and endothelial cell adhesion, and to reduce their protein adhesion. These modifications enable the use of polyurethanes in the manufacture and design of cardiovascular devices.

## 1. Introduction

In the treatment of different cardiovascular pathologies, it is generally necessary to replace the compromised structures and tissues. For decades, biomaterials for cardiovascular applications have been studied and developed to improve their biocompatibility with the human body when implanted, as well as their capability to mimic the behavior of living tissues. Polyurethanes (PUs) are a class of polymers that are used in the preparation of medical devices owing to their biocompatibility, degradability, and non-toxicity [1].

It has been established that PUs are produced by the condensation reaction of an isocyanate and a material with a hydroxyl functionality produced by using either a single type of monomer (homopolymers) or two types of monomers (copolymers). Copolymers are of the following types: random, alternating, segmented, block, and graft [1]. With regard to medical applications, PUs could be used in peritoneal dialysis, neurological leads, and infusion pumps. In addition, PUs could be used as pacemaker leads, vascular grafts, pacemaker lead insulation, probes, wound dressings, feeding tube, cannulas, catheters, or cardiovascular catheters [1].

Microbial colonization is likely to occur on material surfaces in some the PU applications mentioned above. This results in severe and generally life-threatening complications such as infections [1]. Polycarbonate-based PUs, thermoplastic medical-grade PUs, and fluorinated PUs are antimicrobial PUs that can be used in medical applications. Medical-grade PUs are classified into the following types based on hardness: Carbothane^TM^, an aliphatic poly(carbonate)urethane; Tecoflex^TM^, an aliphatic poly(ether)urethane; and Tecothane^TM^, an aromatic poly(ether)urethane [1]. These medical-grade PUs are potential materials for long-term implantation within a living body as “minimally invasive devices” because of their inherent properties, including having a low modulus of elasticity and high ultimate tensile strength [1]. It is established that the synthetic materials used for biomedical applications require remarkable mechanical properties, including blood compatibility, good biostability [2] and hemocompatibility, and the capability to minimize adverse rejection processes mediated by adhering cells or proteins [3].

Scientific publications contain information on the relationship between PU and its use in cardiovascular devices. However, of particular interest to our research group, we seek to develop a systematic review that would enable us to understand why PUs have been used in the manufacture and design of cardiovascular devices, with a focus on the past 15 years.

## 2. Materials and Methods

### 2.1. Review Design

The Preferred Reporting Items for Systematic Reviews and Meta-Analyses (PRISMA) framework was used as the guidelines for this systematic review [4]. This review was not previously registered.

### 2.2. Definition of the Research Question

The two elements format strategy was used for defining the research question. “Cardiovascular devices” and “polyurethane” were the key elements for P (population/phenomena) and O (outcomes), respectively. The research question was formulated as, “Why have PUs been used in the manufacture and design of cardiovascular devices?” This question follows the characteristics of a good research question: the feasible, interesting, novel, ethical, and relevant (FINER) criteria [5].

### 2.3. Eligibility Criteria

For this review, we selected the studies based on the type of article: journal articles and studies concerning the use of PUs in the manufacture and design of cardiovascular devices. We selected books, case reports, case series, conference, not-indexed documents, editorials, letters, manualx, miscellaneous (misc) documents, patents, proceedings, review articles, tech reports, theses, and dissertations as the exclusion criteria.

### 2.4. Information Sources, Search Strategy, and Study Selection

A complete database search was performed by two authors (K.N.-G. and M.F.V.) in November 2019. We used PubMed, Scopus, and Web of Science as the information sources for this review. The search parameters of the study selection were the year, publication state, and language. The search period ranged from 1 January 2005 to 31 December 2019, the publication state was “published”, and the language was limited to English.

The searches in Pubmed, Scopus, and Web of Science were carried out using the search queries [(polyurethane)) AND ((cardiovascular) AND (device))], [(ALL (polyurethane) AND ALL (“cardiovascular devices”))], and [polyurethane AND (cardiovascular AND device)], respectively.

The references were managed with Rayyan Qatar Computing Research Institute (QCRI) [6] to (1) eliminate duplicates and (2) exclude articles with insufficient information (both in the title and in the abstract); articles with keywords such as electrospun, electrospinning or electrospinning technique, drugs, blood vessels, blood contacting, urea, poly(etherurethane urea) (PEUU), poly(carbonate urethane) (PCU), polyhedral oligomeric silesquioxanes (POSS) or simulations; and articles with insufficient information on the use of PU as a material for constructing cardiovascular devices. Specifically, we excluded the articles using electrospun or electrospinning techniques for PU because the use of these techniques has no particular interest to our research group. The inclusion of articles that presented conflicts between two authors was resolved by having the articles evaluated by Luis Eduardo Díaz-Barrera (L.D) and Said Arévalo-Alquichire (S.A).

## 3. Results

### 3.1. Search Results

We used the PRISMA Flow Diagram [4] to summarize the search process (Figure 1). We recovered 1552 articles form the information sources (PubMed, Scopus, and Web of Science).

The searches in PubMed, Scopus, and Web of Science yielded 861, 343, and 348 articles, respectively. In total, 647 articles were excluded after being managed with Rayyan QCRI [6]; 40 full-text articles were assessed for eligibility.

The reasons for the excluded records (n = 647) were: duplicates (n = 205); no information on PU (n = 178); no information on cardiovascular field (n = 81); year of publication (n = 81); no topic (n = 58); insufficient information (n = 37); Publication types: reviews (n = 61); books (n = 52); misc (n = 10); misc and case reports (n = 29); misc and reviews (n = 9); misc and invited commentaries (n = 2); report (n = 2); incorrect publication type (n = 2); misc, state of the art (n = 1); keywords such as electrospinning (n = 22); POSS (n = 12); drug (n = 5); PEUU (n = 5); PCU (n = 2); polyurethane urea (n = 2); blood-contacting (n = 2); poly(ester carbonate urethane) urea (n = 1); non-English (n = 1); and simulations (n = 5).

After a full-text revision, 19 articles were excluded owing to the following reasons: 9 did not contain information on PU characterization and 10 presented PU syntheses using the electrospun or electrospinning technique. A total of 21 papers were included in the analysis.

### 3.2. Characteristics and Results of Included Studies

The following data were tabulated from the included articles: author, year of publication, geographic setting, PU type or chemical composition, type of study (in vitro or in vivo) or type of cells, field of application, and main results (Table 1).

Four of the studies were conducted in the USA [7,8,9,10], three in Italy [11,12,13], two each in India [14,15] and Iran [16,17], and one each in Australia [18], Canada [19], Colombia [20], Germany [3], South Korea [21], and Taiwan [22]. The collaborative studies that were conducted are as follows: Australia–France [23], Brazil–Germany [2]; and People’s Republic of China–Canada [24,25].

Five of the studies used commercial PUs: Elastogran [3], Elastollan^®^ [14], ChronoFlex^®^ [8], Elast-Eon™ [18], and NovoSorb™ [23]. Three of the studies used medical-grade PUs: Tecoflex^®^ [10]; Tecothane™ [7,9].

Seventeen studies conducted in vivo tests with NIH/3T3 mouse fibroblast cell line [11], human umbilical vein endothelial cell (HUVEC) [2,14,16,23,25], L-929 fibroblast [16,20,22]; 3T3 [18,20], H9C2 cardiomyoblasts [12], endothelial colony forming cells (ECFCs) [3], smooth muscle cells (SMCs) [15], A10 SMCs [19], ECs [15], human microvascular endothelial cells (HMEC) [21], cardiomyocytes [13], or hBMSCs [17]. Two studies performed both in vivo and in vitro tests [7,9]. One study conducted only an in vivo test [24]. Two studies conducted neither in vivo nor in vitro tests [8,10]. The studies included in the analysis recommend the use of PUs in cardiovascular devices [22,24], hemodialysis [22], biomedical applications [18,20], blood-contacting devices [3,14], cardiovascular stents [8,15,17,23], heart valves [7,9], heart patches [12], tissue engineering [19,25], and vascular prostheses [21].

All the included studies presented modifications made to PUs with the aim of improving their properties. Six studies presented variations of the chemical compositions of PUs [11,15,16,17,20,24]. Five studies presented modifications to the material surface [7,9,14,21,23] and ten studies described different techniques to prepare PUs [2,3,8,10,12,13,18,19,22,25].

This study had limitations. We only used three databases (PubMed, Scopus, and Web of Science) and only articles in English were screened. The systematic review included articles that represent a small fraction of the literature on synthesis techniques for application in cardiovascular devices.

## 4. Discussion

PUs have found applications in biomedical field because of their properties, i.e., capability of sustaining the contractile forces that originate during the cardiac cycle without undergoing plastic deformation or failure [12], and the capability to imitate the behavior of different tissues [15,20,26,27,28]. The applications of PUs in this field include biostable implants [20] and cardiovascular implantable devices such as cardiac pacemakers [11,18,20], catheters [11,18], prostheses [11,21,29], cardiac assist devices [11], heart patches [12,16], heart valves [11,30] stent [8,15,17,23], and vascular grafts [11,18,31,32].

PUs are thermoplastic elastomers polymers that can be defined as segmented block copolymers, characterized by the presence of two micro-separated phases: soft and hard [11]. This morphology is related to their mechanical properties [8,11,29,33], including their high tensile strength [11], fatigue resistance [34], and elasticity [34]; good tear and abrasion resistance [11]; biodegradability [11]; blood compatibility [34,35]; and biocompatibility [8,11,29,33].

However, studies reported that PUs show severe problems [22] when applied in blood-contacting devices, such as the biodegradation that occurs during long-term implantation [29] by the adhesion of inflammatory cells [27], surface-induced thrombosis [2,22], and protein fouling [22] (which are known to participate in PU biodegradation); and the absence of endothelialization [2].

As a solution to this problem, PUs can be modified in terms of both their chemical composition [11,15,16,17,20,29,36] and surface functionalization to improve their mechanical properties, blood compatibility, and adhesion to ECs, and to reduce protein adhesion [2,3,7,18,24,32,36,37,38,39,40].

### 4.1. PU Modification in Terms of Chemical Composition

Our analysis revealed the use of silicone [11], PCL [12,16,17,19], PCL/PEG [12], castor oil (CO)/aliphatic diisocyanates [20], AgNO_3_, and carbon [15] to modify the chemical compositions.

#### 4.1.1. Silicone

Silicone presents properties such as low surface energy [11]; hemocompatibility [11]; low toxicity [11,29]; remarkable thermal, oxidative, and hydrolytic stability [11]; high flexibility [11]; good biocompatibility [11,29,41]; and long-term biostability [29,41].

#### 4.1.2. PCL

We have identified PCL as a bioresorbable and biocompatible polymer with good mechanical properties [12,16]. PU/PCL blends (copolymer and homopolymer) are well known for their low degradability, cell adhesion, and proliferation, which indicate good biocompatibility. They have been used as materials with elastic memory to achieve self-expansion within the range of the body temperature. Silvestri et al. illustrated that the shape recovery of PU/PCL blends is related to the elastic strain generated during deformation owing to the elasticity of the PU as a matrix phase in these blends [16]. Specifically, PU/PCL blends and PCL/PEG/PCL tri-blocks with different aliphatic and amino-acid-based chain extenders did not present toxicity. The authors recommend the use of the blend for tissue engineering (TE) applications, considering that their mechanical behavior and cell response depend on their chemical composition [12].

#### 4.1.3. CO

A variation of the PU chemical composition with a CO/aliphatic diisocyanates blend is recommended in the included articles. Arévalo et al. used CO as the polyol because of its composition (ricinoleic acid), which presents a structure that enables the synthesis of cross-linked urethanes. Furthermore, it is a renewable source and has low toxicity. The CO/aliphatic diisocyanates (isophorone diisocyanate, IPDI) blend can be used to synthesize biomedical PUs because they do not promote the generation of carcinogenic products, such as the aromatic diamines, in in vivo conditions [20].

#### 4.1.4. Nanomaterial Carbon Dot–Silver Nitrate

Meanwhile, Dura et al. developed a material incorporating a biocompatible nanomaterial carbon dot–silver nitrate (CD-Ag) in a smart polymer matrix for potential use as a stent. The study indicated that they attained a faster self-expansion of the material (in less than a minute), which prevented the migration of the device during in vivo deployment. The authors propose the use of both silver nitrate owing to its potent antibacterial activity, which prevents biofilm formation; and carbon dots owing to their highly polar peripheral groups, which enhance the mechanical properties of the nanocomposite [15].

### 4.2. PU Modification in Terms of Surface Functionalization

Ten studies described how the biomaterials interact with blood. Implantation begins with the blood–foreign material interaction [22,34].

At this stage, the plasma proteins are adsorbed onto the material surfaces, which occurs within a few seconds. This adsorption causes the adhesion of platelets, white blood cells, and a few red blood cells onto the plasma protein layer [34]. Yu et al. revealed that the aggregated platelets on the surface release materials such as adenosine diphosphate (ADP), which results in the formation of thrombin, an insoluble fibrin network, or thrombus [14,22]. Raut et al. explained that the thrombus may obstruct the blood flow and cause device failure. In addition, clots may be released into the systemic blood circulation from the devices that do not fail, resulting in an embolism [14]. Furthermore, Liu et al. highlighted that after the implantation, monocytes may be activated after adhesion to the biomaterials and release cytotoxic mediators, such as cytokines and reactive oxygen species (ROS) [34]. Monocytes have been recognized for their essential role in mediating inflammatory responses [37]. Du et al. described how thrombi that form inside a catheter lumen make the catheter unsuitable for biomedical uses, such as withdrawal of blood, delivery of fluids, or medication. Thrombi, which form outside the catheter, could permanently damage the vessel integrity, resulting in pain and swelling [42].

#### 4.2.1. Surface Functionalization-Modified PU to Promote EC Adhesion and Proliferation

Surface coverage with ECs prevents the material from directly contacting with blood and can be considered as a solution for preventing thrombus formation in cardiovascular implants [2]. ECs are known to provide an antithrombogenic surface by producing antithrombogenic substances [40], such as prostacyclin (PGI2), tissue plasminogen activator (t-PA), heparin-like glycosaminoglycans, and thrombomodulin [21]. The use of ECM proteins such as Coll and Fn as coatings on synthetic polymers enhances endothelialization [23].

Stachelek et al. changed the PU formulation by including cholesterol (Chol). Cholesterol-modified polyurethane (PU-Chol) increased the adhesion of blood outgrowth endothelial cells (BOECs) compared with controls. This change in the adhesion rate is important because it is known that BOECs are an outgrowth of a circulating progenitor cell. These are present in peripheral blood, and thus represent an important potential source of autologous cells for seeding investigations [9].

Heparinized surfaces are clinically used to reduce thrombogenicity [36] as potent anticoagulants that interact strongly with antithrombin (AT) to prevent the formation of fibrin clots [39]. The presence of heparin on the surface positively affects EC growth and proliferation by binding and stabilizing cell growth factors (GFs) [40]. However, a disadvantage of immobilizing heparin onto a polymer surface is that the heparin has low bioactivity [38]. Klement et al. recommended the modification of a PU with an antithrombin–heparin (ATH) covalent complex, which displays the capability for rapid direct inhibition of thrombin [39].

A PU nanocomposite (polymer matrix with embedded nanoparticles of Au-Pt) is a PU modification that has been recommended by Hess et al. to improve cell adhesion, which are functions of the concentration of nanoparticles [3].

The surface functionalization of biomedical devices could be attained by changing the material topographies to enhance the adhesion and growth of ECs [2]. Our results indicate the use of different techniques to enhance surface functionalization, such as direct laser ablation technique (DLA) [2], polymeric endoaortic paving (PEAP) [10], pulsed laser ablation in liquid (PLAL) [3], thermally induced phase separation (TIPS) [13], and freeze-drying [25].

DLA is a technique that modifies the surface topography [2]. Cortella et al. showed the functionalization of PU by DLA-created microtopography, which improved the adhesion and proliferation rate of ECs [2].

Ashton et al. recommends an alternative method to enhance conventional endoaortic therapy to reduce the risk of endoleak. PEAP is a process where a polymer is endovascularly delivered and thermoformed to coat or pave the lumen of the aorta [10]. Meanwhile, the PLAL technique for solid targets is advantageous for the synthesis of biocompatible nanomaterials, because it generates nanoparticles without the need for chemical precursors, which potentially cause cell behavior side effects. Furthermore, the technique enables in situ functionalization with biomolecules and adsorption onto microparticle surfaces [3].

Vozzi et al. recommend the use of TIPS to fabricate oriented scaffolds. This technique modified the PU porosity by modulating the polymer concentration, quenching temperature, thermal gradient, and solvent type [13]. The authors indicate that fiber alignment supports cardiomyocytes in the generation of a tissue-like structure [13]. The last technique identified by us for enhancing surface functionalization was freeze-drying. Jiang et al. employed freeze-drying to develop 3-D porous scaffolds based on the prepared waterborne biodegradable PUs. With this technique, it is possible to obtain scaffolds with an appropriate pore diameter and distribution to enhance the adhesion and proliferation of ECs [25].

#### 4.2.2. Use of Surface Functionalization-Modified PU to Enhance Biocompatibility, Bioactivity, Biodegradation Resistance, and Electrical Conductivity

Thampi et al. and Cortella et al. used surface functionalization with antithrombotic agents or immobilization of molecules, such as polyethylene oxide (PEO), heparin, albumin, and chitosan (CS), to improve the blood compatibility of biomedical devices [2,43]. Roth et al. and Yu et al. recommended the application of biologically active substances, such as anticoagulants [14], fibrinolytic enzymes or proteins [14], heparin [22], CS [22], and dextran sulfate (DS) [22], as surface modification substances [14,22] in order to enhance the compatibility of PUs with the blood components.

Another PU modification technique was presented by Gu et al. through the incorporation of PEG to enhance bioactivity. PEG chains were selected as hydrophilic and flexible spacers between the biological molecules and PU backbones [24]. Hess et al. recommended a PU modification with embedded nanoparticles of Au-Pt to improve biocompatibility, which are functions of the concentrations of nanoparticles [3]. Stachelek et al. recommended the use of modified PU with DBP to provide antioxidant activity to prevent biodegradation. The microscopy results showed that DBP-modified PU confers biodegradation resistance to surface cracking with dose-dependent DBP loading [7].

A different PU modification was presented for electrically conductive polymeric materials. These materials could be used in biomedical applications, such as for biosensors, drug delivery systems, biomedical implants, and TE. Kaur et al. recommended the incorporation of conductive fillers such as graphene to enhance the electrical conductivity without changing the polymeric characteristics [18].

## 5. Conclusions

PUs have found applications in medical devices because of their properties—their capability to tolerate contractile forces that originate during the cardiac cycle without undergoing plastic deformation or failure, and their ability to imitate the behaviors of different tissues. Although studies have reported that PUs suffer from severe problems when applied in blood-contacting devices that are implanted for long periods, they can be modified in terms of both the chemical composition and surface characteristics to improve their mechanical properties (blood compatibility and EC adhesion) and reduce protein adhesion. These modifications enable the use of PUs in the manufacture and design of cardiovascular devices.

Our analysis revealed the following:For stent design, it is important that the selected material displays self-expandable and shape memory behavior, which must be maintained at temperatures similar to those encountered in the human body. In addition, this expansion must be carried out as quickly as possible to prevent the migration of the stent during surgery. Modified PUs, such as those with added PCL or carbon dot–silver nanohybrid, show high modulus and tensile strength with low elongation and biocompatibility. Furthermore, these maintain self-expandable and shape memory behaviors. All these properties permit us to propose the use of these PUs as potential materials for stent implants.Among cardiovascular devices, heart patches and heart valves require the use of materials with appropriate properties such as strength and an elastomeric mechanical behavior to tolerate the contractile cardiac tissue and support its regeneration. For the design of heart patches and heart valves, it is important to create a structure that is similar to the muscle tissue. PUs are appropriate materials for cardiac applications because their biocompatibility and elastomeric behavior enable them to resist the cyclic heart stresses without deformation or failure. The PU structure can be modified to create anisotropic microstructures that may mimic the heart tissue function of different pore sizes. This could promote cell colonization, cell migration, nutrient supply, and vascularization.In general, for blood-contacting devices, the interactions between the material and blood generate cell responses that could favor the formation of thrombi. PUs provide a surface that can be modified to reduce non-specific protein adsorption and promote endothelial cell attachment and proliferation. This can enhance the biocompatibility and hemocompatibility. This implies that the fabrication of blood-contacting devices with modified PUs could decrease the numbers of repeated operations and deaths in cardiovascular surgery [3].

## Figures and Tables

**Figure 1 materials-13-03250-f001:**
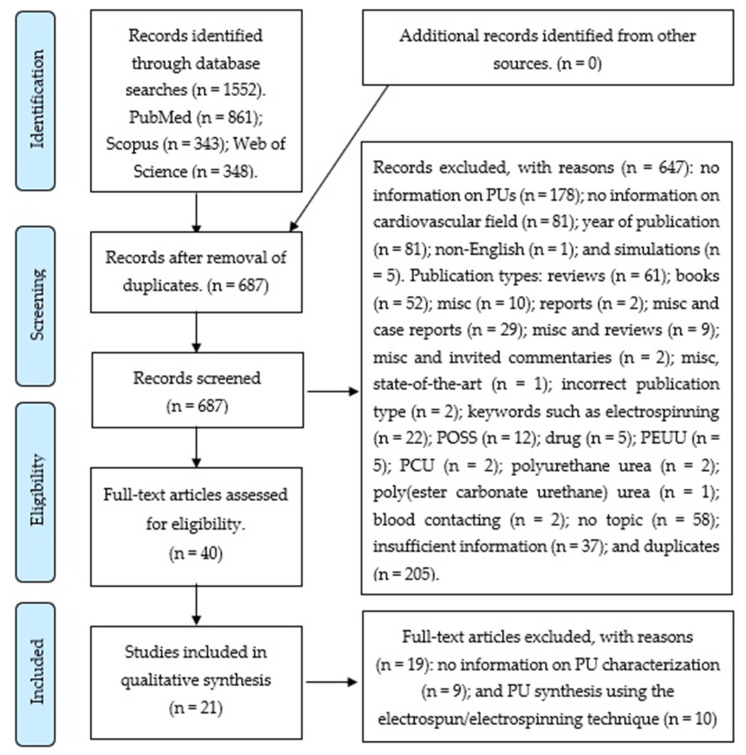
Preferred Reporting Items for Systematic Reviews and Meta-Analyses (PRISMA) flow diagram.

**Table 1 materials-13-03250-t001:** Characteristics and results of Included Studies.

Author	Year of Publication	Geographic Setting	PU/Chemical Composition	Type of Study (in vitro, in vivo)/Cells	Field of Application	Main Results (Extracted and/or Adapted)
Trigwell et al. [8]	2005	USA	ChronoFlex^®^ AR PU/polytetrafluoroethylene (PTFE)	x ^1^	Cardiovascular stents	A method of encasing cardiovascular stents with an expandable PU coating was developed to provide a smooth homogeneous inner wall that allows for a confluent growth of endothelial cells. The PU film covered the metal wire stent structure, thereby minimizing biocorrosion of the metal (stainless steel or nitinol) and providing a homogeneous surface. The stent structure covered with a film of less than 25 µm could display sufficient corrosion resistance and flexibility without producing excess stress in the structure.
Cho et al. [21]	2005	South Korea	Poly (oxytetramethylene) glycol (PTMEG)/ 1,4-butane diol (BD)/4,4′-methylenediphenyl diisocyanate (MDI) + Collagen (Coll)	In vitro. HMEC	Vascular prostheses	The semi-microporous segmented polyurethane (SPEU) used in this work showed properties that promote endothelial cell attachment and proliferation. The endothelial cells were attached to the SPEU semi-pores, which resulted in less platelet adhesion. The authors recommend collagen as a coating. However, a collagen coating on the SPEU surface affected the endothelial cell attachment, as well as platelet attachment. It is not recommended for use if factors such as blood coagulation and prosthesis patency are important.
Yu et al. [22]	2006	Taiwan	Thermoplastic polyurethane (TPU)/chitosan (CS)/dextran sulfate (DS)	In vitro. L-929 fibroblast	Antithrombo-genic coating for application in hemodialysis or cardiovascular devices.	In this work, PEM of CS/DS (CS as a positively charged agent, and DS as a negatively-charged and an antiadhesive agent) were deposited onto the aminolyzed TPU film surface by the LBL self-assembly technique. The authors note that the deposition of over four bilayers with DS, as the outermost layer could improve the hydrophilicity of the TPU film, suppress the protein adsorption and platelet adhesion, and prolong the blood coagulation time.
Stachelek et al. [9]	2006	USA	Tecothane™/cholesterol (Chol)	In vitro, in vivo. Sheep blood outgrowth endothelial cells (BOECs)	Heart valve	PU-Chol has been presented as an option for BOECs for applications such as PU heart valve leaflet implants. This work indicated that PU-Chol has significantly better BOEC adhesion properties than unmodified PU under simulated and in vivo heart valve shear force conditions.
Stachelek et al. [7]	2007	USA	Tecothane™/di-tert-butylphenol (DBP)	In vitro, in vivo. THP-1 cells	Heart valve leaflets and PU artificial heart devices.	This work showed that the covalent modification of PUs such as Tecothane by using DBP conferred biodegradation resistance in vivo and that this biodegradation is dependent on the DBP dose. It is important to note that the modification with DBP could be effective in trapping oxygen radicals that are released from adherent MDMs that interact with PUs.
Ajili et al. [17]	2009	Iran	Polyesterurethane (MDI/1,4-butanediol (BDO)/polycaprolactone (PCL)	In vitro. human bone marrow mesenchymal stem cells (hBMSCs)	Cardiovascular stents	In this work, authors observed that shape memory materials have been proposed for cardiovascular stents owing to their self-expansion capability. This capability is important for polymeric stent deployment at temperatures near the body temperature. To work on this capability, the investigators used crystallinity-induced shape memory effect to incorporate elastic memory in a stent. They used PU/PCL blends as materials for shape memory stents. The PU/PCL blend compositions and crystallization conditions were modified. The PU/PCL (70/30) blend showed remarkable biocompatibility, which was indicated by the adhesion and proliferation of bone marrow mesenchymal stem cells compared with the other blends. Furthermore, this blend is a potential material for use in stent implantx.
Sharifpoor et al. [19]	2010	Canada	D-PHI. 2 porogen (sodium bicarbonate (salt) and poly(ethylene glycol) (PEG))	In vitro. A10 smooth muscle cells (SMCs) -thoracic aorta of embryonic rats.	Vascular tissue engineering applications	In this study, the authors used double porogen (PEG–salt) to optimize the pore interconnectivity and to increase the porosity in the PU without compromising on the scaffold mechanical integrity. The materials were tested under dynamic mechanical stretching to mimic the biomechanical conditions. The use of PEG–salt porogens was effective in improving the pore interconnectivity through the production of micropores in the range of 1 to −5 µm, and in increasing the total scaffold porosity by enabling the addition of more salt within the monomer–porogen mixture.
Ashton et al. [10]	2010	USA	PCL/Tecoflex^®^ SG-80A	x	Cardiovascular applications. Polymeric endoaortic paving	In this work, PCL/PU blends were used as paving materials for PEAP. The authors noted that the blends’ stiffness was similar to that of aortic tissue, depends on the PCL content, and may be affected by thermoforming and degradation. The PEAP, consisting of a PCL/PU blend, may be effective for developing highly advanced endoaortic therapy.
Silvestri et al. [11]	2011	Italy	Poly(dimethyl siloxane)—poly (tetramethylene oxide) (PDMS–PTMO) -based PU. Clay as a filler	In vitro. NIH 3T3 mouse fibroblast	Cardiovascular devices	In this work, the authors modified the PU chemical composition and used clay as a filler. They described that these modifications enable the determination of an appropriate formulation for biostable cardiovascular devices. Among the chemical compositions, the authors used a nonaromatic diisocyanate (HDI). This enables the development of mechanical properties close to those of the native mitralic tissue and prevents the production of highly toxic aromatic diamines. The authors indicated that a higher percentage of PTMO in the soft segment improved the mechanical performances of PUs. That is, it increased the Young’s modulus, the stress at break, and the maximum strain. The Young’s modulus values at 37 °C were included in the required range (6 MPa in the circumferential (parallel to the annulus) direction and 2 MPa in the longitudinal (perpendicular to the annulus) direction) for annuloplastic applications.
Jiang et al. [25]	2012	People’s Republic of China–Canada	Biodegradable waterborne polyurethane (WBPU): isophorone diisocyanate (IPDI) /BDO/PEG/PCL/L-lysine	In vitro. HUVECs	Soft tissue engineering	In this study, the freeze-drying technique was used to fabricate 3-D interconnected porous scaffolds using a non-toxic, waterborne, biodegradable PU emulsion. The goal was to prepare scaffolds with appropriate pore diameter, pore diameter distribution, and porosity for use in soft tissue engineering. The authors observed that the relatively smaller pore diameter, narrower pore diameter distribution, and lower porosity were more advantageous for the scaffold endothelialization.
Silvestri et al. [12]	2013	Italy	PEG/PCL/1,4-diisocyanatobutane (BDI). L-Lysine Ethyl Ester and AAK as chain extenders.	In vitro. Rat heart cell line (H9C2 cardiomyoblasts)	Heart patches for myocardial function restoration and cardiac tissue regeneration after an MI	The authors used TIPS technique to prepare scaffolds with an analogous structure of the striated myocardial tissue for myocardial applications. The authors noted that the material was similar to the streaked muscle tissue because of the presence of the anisotropic microstructure. Thereby, the so called “structural biomimicry” was achieved, which is a requirement for the application of biomaterials in TE. The different pore sizes can promote different processes: large pores favor cell colonization, cell migration, and nutrient supply; meanwhile, small pores can promote vascularization.
Hess et al. [3]	2013	Germany	Elastogran/Au-Pt	In vitro. ECFCs	Blood-contacting medical devices	The authors applied the PLAL technique to generate TPU–noble metal nanocomposites with different concentrations of Pt or Au nanoparticles between 0 and 1 wt%. The presence of metal nanocomposites in TPU improved the biocompatibility and cell adhesion. The authors relate this effect to the hydrophilic and negatively-charged surface. Results showed that ECFCs seeded onto the nanocomposites remained in a nonthrombogenic and noninflammatory state. Thereby, the material can potentially decrease the number of reoperations and deaths in the field of cardiovascular medicine.
Baheiraei et al. [16]	2014	Iran	PEG/PCL/IPDI/aniline	In vitro. L-929 HUVECs	Cardiac tissue engineering	The authors worked on materials for TE. They studied the low conductivity of the patch in cardiac TE because this characteristic could limit the patch’s capability to couple transplanted cells electrically to the local host myocardium. This study recommends the use of oligoaniline as an electroactive conductive polymer. These materials were non-toxic, supported cell proliferation and attachment, and combined with antioxidant properties.
Sgarioto et al. [23]	2014	Australia, France	NovoSorb™. extracellular matrix (ECM) as Coll and fibronectin (Fn)	In vitro. HUVECs	Cardiovascular stents	The authors studied materials for application in cardiovascular stents. Specifically, they worked with biodegradable PUs with different HS percentages. The main results revealed that the PUs showed high modulus and tensile strength with low elongation, which are key characteristic for fabricating vascular stents. The tensile strength reduced significantly upon gamma sterilization in the case of PUs with a low content of HSs. Meanwhile, the strength was maintained for the materials with high HS contents.
Gu et al. [24]	2015	People’s Republic of China–Canada	Polycarbonate urethanes with poly (ethylene glycol) diglycidyl ether (PEG-PO) or poly (ethylene glycol) bis(amine) (PEG-PN)	In vivo. Sprague-Dawley (SD) rats	Coatings for cardiovascular devices	The authors aimed to study the toxicity development in cardiovascular implants. The materials synthesized included hydrophilic PEG side-chains attached to the HS. The authors noted that these chains increase the hydrophilicity of the macromolecules and modify the hydrogen bonds among the HS, both of which favor hydrolytic degradation. The results did not show any maternal and fetal toxicity. The use of this material in cardiovascular applications was proposed based on this observation.
Kaur et al. [18]	2015	Australia	Graphene/ElastEon™ composite films	In vitro. L-929 cells	Biomedical applications	The authors used three methods (solution mixing, melt processing, and in situ methods) to synthesize conductive composites of a siloxane PU and graphene for potential use in biomedical applications. The results showed that the solution mixing method yielded composites with the highest electrical conductivity and that it is suitable for preparing composites with better mechanical properties. The authors also noted that the materials were not cytotoxic to living cells in vitro and are potentially useful in biomedical applications.
Arevalo et al. [20]	2016	Colombia	Castor oil/PCL/IPDI/CS	In vitro. L-929 and 3T3 cells	Biomedical applications related to soft and cardiovascular tissues	The authors aimed to study materials for potential biomedical applications. They synthesized PU from castor oil, PCL, and IPDI, using CS as the additive. The results showed that the presence of CS in materials enhances the ultimate tensile strength and does not affect the strain at fracture in PUs with 5% *w*/*w* of PCL and CS in the range of 0–2% *w*/*w*. The authors noted that PUs had mechanical properties similar to those of the aorta and skin.
Duarah et al. [15]	2016	India	PCL/2, 4-2, 6-toluene diisocyanate (TDI)/silver nitrate (AgNO_3_) /soluble starch	In vitro. SMCs. endothelial cells (ECs).	Cardiovascular stents	The goal of this work was to synthesize materials for rapid self-expandable stents for possible endoscopic surgeries. The authors used a one-pot single step technique to obtain a carbon dot–silver nanohybrid. The results showed that it was possible to obtain a material with self-expandable and shape memory behaviors, as well as enhanced thermal and mechanical properties.
Cortella et al. [2]	2017	Brazil–Germany	CO/MDI	In vitro. iHUVEC	Cardiovascular devices	In this work, the authors recommended the use of the DLA technique for producing patterned topographies in the micrometer range. The results show that the topographical patterns produced by the DLA technique on cellular processes may contribute to the development and maintenance of a functional endothelium on target surfaces by the functionalization of their blood-contacting surfaces.
Vozzi et al. [13]	2018	Italy	PCL/BDI/L-lysine ethyl ester	In vitro. Cardiomyocytes	Cardiac tissue engineering	The authors aimed to design a material that mimics cardiac tissue properties. They used the TIPS technique to fabricate scaffolds, which were functionalized with fibronectin. The results highlighted the feasibility of fabricating PU scaffolds with porous-aligned structures and mechanical properties consistent with those of the myocardial tissue.
Rau et al. [14]	2019	India	Elastollan/PEG/gelatin as a surface modifier	In vitro. HUVECs	Blood-contacting devices.	The authors recommended PU modification with PEG for blood-contacting applications. They observed that the PU modification reduced non-specific protein adsorption and promoted endothelial cell attachment and proliferation. They also emphasized that the PU surfaces modified with PEG and gelatin enhanced the hydrophilicity. This yielded enhanced biocompatibility and hemocompatibility.

^1^ This information is not reported in the article.

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
