# Peer review of "Why Polyurethanes Have Been Used in the Manufacture and Design of Cardiovascular Devices: A Systematic Review"

_materials, 2020, doi:10.3390/ma13153250_

Round 1

Reviewer 1 Report

Well written 

Author Response

Reviewer 1

Comment and/or suggestion

Authors answer

Well written

Thanks for your comment, we really appreciated.

Reviewer 2 Report

The goal of this review article is to evaluate the use of PU in the manufacture and development for cardiovascular applications from the final 21 articles from 2005-2019. The authors have summarized and concluded that PUs are used for cardiovascular devices due to their capability to tolerate contractile forces that originate during the cardiac cycle without undergoing plastic deformation or failure, and the ability to imitate the behavior of different tissues. PUs can be chemically modified and functionalized to provide appropriate properties to improve EC adhesion, reduce protein adhesion, and blood compatibility. The methods and results are well written. However, there are still unclear and inappropriate explanations throughout the article.

  1. The introduction is lacking essential information. The authors should elaborate the key physical, mechanical, and biological properties of PU in the aspect of cardiovascular application (i.e, providing the number of mechanical properties)
  2. In the introduction, the authors mentioned that “PUs are potential materials for long-term implantation within a living body as “minimally invasive devices” because of their properties, including low modulus of elasticity and high ultimate tensile strength”.  This isn't very clear. Is that true that PU has low elastic modulus (which is usually 15.1–151.4 MPa)? Are mechanical properties the only reason that makes PU as a candidate for minimally invasive devices? Please revise this statement and make it clear.
  1. In the introduction, the authors mentioned that “Polycarbonate-based PU, thermoplastic medical-grade PU, and fluorinated PU are a few of the materials that can prevent infections”. It is not true based on the way it wrote. There are many bioengineering strategies currently used to reduce or prevent infection of implants. Did the authors try to compare only with PU derivatives? If so, please revise.
  2. In the method, why the authors exclude the articles using electrospun/electrospinning techniques for PU? Please clarify in the method section/ results
  3. In the discussion section, the authors should categorize into a subsection to make it clear and easy to understand. It is unorganized in the way it wrote. For instance, the authors can categorize the discussion based on the modified properties: 1) modified PU to promote EC adhesion and proliferation, 2) modified PU to enhance biocompatibility and bioactivity, or categorize based on modified techniques (chemical modification, physical modification, nanomaterial functionalization). 

Author Response

Reviewer 2

Comment and/or suggestion

Authors answer

1. The introduction is lacking essential information. The authors should elaborate the key physical, mechanical, and biological properties of PU in the aspect of cardiovascular application (i.e, providing the number of mechanical properties)

Thanks for your comment. We improved the introduction and made some changes to make it clear.

2. In the introduction, the authors mentioned that “PUs are potential materials for long-term implantation within a living body as “minimally invasive devices” because of their properties, including low modulus of elasticity and high ultimate tensile strength”.  This isn't very clear. Is that true that PU has low elastic modulus (which is usually 15.1–151.4 MPa)? Are mechanical properties the only reason that makes PU as a candidate for minimally invasive devices? Please revise this statement and make it clear.

Thanks for your comment. We made a re-read of our introduction, we checked the information. We made some changes to make it clear.

Is that true that PU has low elastic modulus (which is usually 15.1–151.4 MPa)?

When we expressed "PUs are potential materials for long-term implantation within a living body as “minimally invasive devices” because of their properties including low modulus of elasticity and high ultimate tensile strength" we use this sentence in a general way, however this information relates to Medical-Grade Polyurethane. Accordingly, we placed the sentence in the Medical-Grade Polyurethane descriptions.

Are mechanical properties the only reason that makes PU as a candidate for minimally invasive devices?

No, it is established that the synthetic materials used for biomedical applications require remarkable mechanical properties including blood compatibility, good biostability and hemocompatibility, and the capability to minimize adverse rejection processes mediated by adhering cells or proteins. We improved the introduction and made some changes to make it clear.

3. In the introduction, the authors mentioned that “Polycarbonate-based PU, thermoplastic medical-grade PU, and fluorinated PU are a few of the materials that can prevent infections”. It is not true based on the way it wrote. There are many bioengineering strategies currently used to reduce or prevent infection of implants. Did the authors try to compare only with PU derivatives? If so, please revise.

We did a re-read of our introduction, we checked the information. We made some changes to make it clear.

We expressed “Polycarbonate-based PU, thermoplastic medical-grade PU, and fluorinated PU are a few of the materials that can prevent infections”. We use this sentence in a general way, however, this information relates to antimicrobial PUs that can be used in medical applications. Therefore, we made some changes to make it clear.

There are many bioengineering strategies currently used to reduce or prevent infection of implants. Did the authors try to compare only with PU derivatives?

There are indeed many Bioengineering strategies that can prevent infections, but these are not evaluated in this article; in our study we focus on Polyurethanes.

4. In the method, why the authors exclude the articles using electrospun / electrospinning techniques for PU? Please clarify in the method section/ results

Thanks for your comment. We seek to develop a systematic review that would enable us to understand why PUs have been used in the manufacture and design of cardiovascular devices, with a focus on the past 15 years.

However, we exclude the articles using electrospun / electrospinning techniques for PU because the use of these techniques has not particular interest to our research group. We made some changes to make it clear.

5. In the discussion section, the authors should categorize into a subsection to make it clear and easy to understand. It is unorganized in the way it wrote. For instance, the authors can categorize the discussion based on the modified properties: 1) modified PU to promote EC adhesion and proliferation, 2) modified PU to enhance biocompatibility and bioactivity, or categorize based on modified techniques (chemical modification, physical modification, nanomaterial functionalization).

Thanks for your comment. We made some changes at discussion section to make it clear. We established a subsection based on PU modification in terms of chemical composition and surface functionalization.

Reviewer 3 Report

This work focuses on the use of polyurethanes (PU) in the manufacture and design of cardiovascular devices. The authors ran a good database using PubMed, Scopus and Web of Science as sources of information. The manuscript addresses an interesting topic, moreover, the paper also uses good English. Therefore, it may be recommended for publication after minor revision:

  1. In my opinion, the introduction is too short, the authors should add much more information about polyurethane such as the advantages and limitations of the use of this material, the reasons why it is used in the manufacture of cardiovascular devices, and also, it is used also for other medical devices?
  2. Discussion Section, pag 10 line 266: Authors should better describe the techniques to enhance surface functionalization of the materials, also comparing the different techniques and explain what is the best technique to modify the surface of the materials for the cardiovascular devices

Author Response

Reviewer 3

Comment and/or suggestion

Authors answer

1. The authors should strive to better explain why such a high number of studies were excluded.

Thanks for your comment. The high number of excluded articles is presented when integrating the three information sources (PubMed, Scopus, and Web of Science), which generates: 205 duplicate articles that correspond to 32%, and articles that do not present information on PU in the title and / or in the summary, which corresponds to 27 %. We made some changes at the results section to make it clear.

2. Could you please specify what do you understand by "irrelevant year of publication"? Was it outside your search targets (2005-2019)?

Thanks for your comment. Grammar mistake. We wanted to express that these numbers of articles were outside our search target (2005-2019). We change it to make it clear.

3. There were no studies that analysed the wear of the polyurethane surfaces by microscope in artificial hearts/valves?

Thanks for your comment. In our work, we find articles that use microscopy techniques to analyze the surface of the material. Some Authors use these techniques to analyze EC adhesion and proliferation; while others, associate the results with biodegradability. We made some changes at discussion section (section 3.2.2.) to make it clear

Reviewer 4 Report

1) The authors should strive to better explain why such a high number of studies were excluded.

2) Could you please specify what do you understand by "irrelevant year of publication"? Was it outside your search targets (2005-2019)?

3) There were no studies that analysed the wear of the polyurethane surfaces by microscope in artificial hearts/valves?

Author Response

PRISMA 2009 Checklist

Round 2

Reviewer 2 Report

The authors have revised the manuscript thoroughly to address all the concerns. The manuscript is well written and organized.